# D$^2$TV: Dual Knowledge Distillation and Target-oriented Vision Modeling for Many-to-Many Multimodal Summarization

**Yunlong Liang[1]\*, Fandong Meng[2], Jiaan Wang[3], Jinan Xu[1]†, Yufeng Chen[1] and Jie Zhou[2]**

[1]Beijing Key Lab of Traffic Data Analysis and Mining,
Beijing Jiaotong University, Beijing, China
[2]Pattern Recognition Center, WeChat AI, Tencent Inc, China
[3]School of Computer Science and Technology, Soochow University, Suzhou, China
{yunlongliang,jaxu,chenyf}@bjtu.edu.cn
{fandongmeng,withtomzhou}@tencent.com    jawang.nlp@gmail.com

## Abstract

Many-to-many multimodal summarization (M$^3$S) task aims to generate summaries in any language with document inputs in any language and the corresponding image sequence, which essentially comprises multimodal monolingual summarization (MMS) and multimodal cross-lingual summarization (MXLS) tasks. Although much work has been devoted to either MMS or MXLS and has obtained increasing attention in recent years, little research pays attention to the M$^3$S task. Besides, existing studies mainly focus on 1) utilizing MMS to enhance MXLS via knowledge distillation without considering the performance of MMS or 2) improving MMS models by filtering summary-unrelated visual features with implicit learning or explicitly complex training objectives. In this paper, we first introduce a general and practical task, *i.e.*, M$^3$S. Further, we propose a dual knowledge distillation and target-oriented vision modeling framework for the M$^3$S task. Specifically, the dual knowledge distillation method guarantees that the knowledge of MMS and MXLS can be transferred to each other and thus mutually prompt both of them. To offer target-oriented visual features, a simple yet effective target-oriented contrastive objective is designed and responsible for discarding needless visual information. Extensive experiments on the many-to-many setting show the effectiveness of the proposed approach. Additionally, we will contribute a many-to-many multimodal summarization (M$^3$Sum) dataset.[1]

## 1 Introduction

Given a document input in the source language (*e.g.*, English) and its corresponding image sequence, multimodal monolingual summarization (MMS) aims to generate a summary in the same

---

**Article:**[...]One worry is that this would trigger panicked reactions from domestic investors and lead to a stock market crash.[...] China's central bank has been repeatedly propping up the stock market to ensure stability.[...]. But that did not happen - causing panic to ripple out and a dramatic drop in shares on Monday. One of the possible triggers for the drop in past trading sessions was the earlier decision by the central bank to devalue the yuan and allow it to trade more flexibly. [...], sending a first wave of insecurity through markets. China's stock market slump caused investor uncertainty to spread across the region and then around the globe, destabilising stock markets in New York and Europe. [...]But now the tables have turned, he says. "The global economy and global markets have a 'Made in China' label on them." Monday's global turmoil sparked fears of another international financial meltdown. They are however warning of further slumps in the long run. [...]. The turmoil caused by China's stock slump "suggested that the great unwind of the excesses is beckoning". [...]

**(MMS) English Summary:** The repercussions from "Black Monday" - the global markets turmoil caused by a plunge in Chinese stocks - continue to be felt...

**(MXLS) Russian Summary:** Влияние « черного понедельника» - глобальных рыночных потрясений, вызванных падением китайского фондового рынка - продолжало проявляться во вторник.

**(MXLS) Urdu Summary:** چین کے ذریعے ایک دھوپ کی وجہ سے ہوتی ہے، سورج کے دن بھی احساس کیا جاتا ہے.

**(MXLS) Indonesian Summary:** Konsekusi dari "Senin Hitam" - gangguan pasar global disebabkan oleh penurunan saham Cina - terus terasa pada Selasa.

Figure 1: An example of our M$^3$Sum dataset. Inputs: an article and corresponding image sequence; Output: summaries in different languages. MMS: the summary in the same language as the input article; MXLS: the summary in a different language from the input article. The M$^3$S setting covers both MMS and MXLS.

---

language (*i.e.*, English) while the goal of multimodal cross-lingual summarization (MXLS) is to produce a summary in a different language (*e.g.*, Chinese). With the rapid increase of multimedia data, the MMS (Tjondronegoro et al., 2011; Evangelopoulos et al., 2013; Erol et al., 2003; Li et al., 2017, 2018a; Sanabria et al., 2018; Zhu et al., 2018; Chen and Zhuge, 2018; Li et al., 2020a; Fu et al., 2021; Zhao et al., 2022) and MXLS (Liu et al., 2022) tasks have attracted much attention in the research community because both tasks can help users quickly master the core idea from the cumbersome multimodal data. Essentially, the many-to-many multimodal summarization (M$^3$S) consists of MMS and MXLS tasks, which generate summaries in any language given the multimodal inputs (in any language), as Fig. 1 shows. Intuitively, the many-to-many setup should be more general and

---

\*Work was done when Liang and Wang was interning at Pattern Recognition Center, WeChat AI, Tencent Inc, China.

†Jinan Xu is the corresponding author.

[1]The code and data are publicly available at https://github.com/XL2248/D2TV.

practical for its application in the multilingual and multimodal world (Wang et al., 2023b).

In the literature, although plenty of studies have been carried out on MMS or MXLS, there is only one study that involves both of them, *i.e.*, Liu et al. (2022) devise a triple-stage training framework and distill the knowledge from MMS to enhance MXLS while ignoring the performance of MMS. Despite their effectiveness on MXLS, to our knowledge, little research attention has been paid to simultaneously supporting both MMS and MXLS tasks and prompting both of them. Besides, the visual features generally include noise which is summary-unrelated. Thus the remaining work mainly focuses on improving MMS models by filtering these noises with (a) implicit learning or (b) complex training objectives. For (a), researchers design various fusion methods to effectively model the interactions between textual articles and visual features (Liu et al., 2020; Yu et al., 2021; Palaskar et al., 2019; Zhang et al., 2021a). For (b), to explicitly filter needless visual information, Liang et al. (2022b) present two well-designed auxiliary tasks, *i.e.*, vision to summary and masked image modeling. Albeit effective, implicit learning via the MMS objective may limit the potential of visual features, and explicit training objectives are complex and time-consuming to be trained and applied in the real world.

To address these issues, in this paper, we first introduce a more general task, *i.e.*, M³S, which supports both MMS and MXLS tasks. Further, we propose a **D**ual knowledge **D**istillation and **T**arget-oriented **V**ision enhanced framework, named D²TV, for the new task. Specifically, the dual knowledge distillation approach ensures that the knowledge from MMS can be transferred to MXLS and vice versa, and thus mutually improve both tasks. Furthermore, to discard the summary-unrelated visual information, a target-oriented contrastive objective is devised to directly optimize the visual features. In this way, the model is enhanced to explicitly exploit the summary-oriented visual features, thereby yielding more accurate summaries.

To validate the D²TV framework, we provide a **M**any-to-**M**any **M**ultimodal **Sum**marization (M³Sum) benchmark dataset by reorganizing the cross-lingual summarization dataset (Bhattacharjee et al., 2022) and MM-Sum dataset (Liang et al., 2022b). The M³Sum covers 44 languages and thus involves 44*44 language directions. To efficiently evaluate our approach, we randomly select 4 languages (*i.e.*, English, Indonesian, Russian, and Urdu[2]), which consist of 4*4 language directions. We implement our approach grounding on two generative pre-trained language models, *i.e.*, mT5 (Xue et al., 2021) and mBART-50 (Tang et al., 2021). Extensive experiments on both backbones show that our model significantly outperforms related methods in terms of ROUGE (Lin, 2004) and BERTScore (Zhang et al., 2020) scores, demonstrating its effectiveness. The human evaluation further suggests the superiority of our approach. In summary, our main contributions are:

- To the best of our knowledge, we are the first that introduces the general many-to-many multimodal summarization (M³S) task and contributes a corresponding benchmark dataset.
- We propose a dual knowledge distillation and target-oriented vision modeling framework for the M³S task.
- Experiments on M³Sum benchmark show that our model builds new state-of-the-art performance, showing the effectiveness of the proposed approach.

## 2 Background

### 2.1 Problem Formulation

Given an input article $\mathcal{X}^{L_1}=\{x_k^{L_1}\}_{k=1}^M$ in language $L_1$ and its corresponding visual features $\mathcal{V}=\{v_{ij}\}_{i=1,j=1}^{i\leq n,j\leq m}$, where $x_k^{L_1}$ denotes the $k$-th token, and $M$ is the number of tokens in the article, and $v_{ij}$ represents the detected $j$-th object of the $i$-th image ($n$, $m$ is the number of images and detected objects in each image, respectively), the many-to-many multimodal summarization task is defined as:

$$p(\mathcal{Y}^{L_2}|\mathcal{X}^{L_1},\mathcal{V}) = \prod_{t=1}^N p(y_t^{L_2}|\mathcal{X}^{L_1},\mathcal{V},y_{<t}^{L_2}),$$

where $y_{<t}^{L_2}$ indicates the tokens before the $t$-th time step of the summary $\mathcal{Y}^{L_2}=\{y_t^{L_2}\}_{t=1}^N$ in language $L_2$ and $N$ is the number of tokens in the summary. The $L_1$ and $L_2$ can be any language.

### 2.2 The MMS Model

Following Yu et al. (2021); Liang et al. (2022b), the MMS model is an extension of the pre-trained language model (*e.g.*, mT5 (Xue et al., 2021)) based

---

[2]Urdu is the low-resource language.

on Transformer architecture (Vaswani et al., 2017). As shown in the left part of Fig. 2, it includes four modules: textual encoder, visual encoder, text-vision fusion, and decoder.

**Textual Encoder.** The textual encoder consists of $N_e$ stacked layers, where each layer consists of two sub-layers, a multihead self-attention sub-layer (SelfAttn) and a position-wise feed-forward network (FFN) sub-layer:

$$\mathbf{S}_T^\ell = \text{SelfAttn}(\mathbf{H}_T^{\ell-1}) + \mathbf{H}_T^{\ell-1}, \ \mathbf{S}_T^\ell \in \mathbb{R}^{M \times d},$$
$$\mathbf{H}_T^\ell = \text{FFN}(\mathbf{S}_T^\ell) + \mathbf{S}_T^\ell, \ \mathbf{H}_T^\ell \in \mathbb{R}^{M \times d},$$

where $\mathbf{H}_T^{\ell-1}$ and $\mathbf{H}_T^\ell$ denote the inputs and outputs of the $\ell$-th encoder layer, respectively, and $\mathbf{H}_T^0$ is initialized as the embedding of input tokens $\mathcal{X}^{L_1}$ and $d$ is the hidden dimension.

**Visual Encoder.** Following Yu et al. (2021); Liang et al. (2021, 2022c,a); Zhang et al. (2021a,b), the visual encoder is also the Transformer (Vaswani et al., 2017) encoder with $N_v$ stacked layers. The difference is the visual inputs. Generally, there is an image sequence to be extracted by the Faster R-CNNs (Ren et al., 2015) pre-trained on Visual Genome (Krishna et al., 2017). Specifically, for the $i$-th input image, we obtain a set of detected objects from Faster R-CNNs, *i.e.*, $\mathbf{I}_i = \{\mathbf{o}_{i,1}, \mathbf{o}_{i,2}, \mathbf{o}_{i,3}, ..., \mathbf{o}_{i,m}\}$, where $m$ is the number of extracted objects and $\mathbf{o}_{i,*} \in \mathbb{R}^{d_v}$. Each object is captured by a dense feature representation, which can be mapped back to a bounding box / region (*i.e.*, Region-of-Interest (RoI)). Finally, the image sequence is converted to visual features $\mathbf{I} = \{\mathbf{o}_{ij}\}_{i=1,j=1}^{i \le n, j \le m}$. Following Cho et al. (2021), the RoI bounding box coordinates $\mathbf{E}_{ij}^{box}$, image id embedding $\mathbf{E}_i^{img}$, and region id embedding $\mathbf{E}_j^{reg}$ are added on the visual features to keep the order information of the image sequence:

$$\mathbf{v}_{ij} = \mathbf{o}_{ij} + \mathbf{E}_{ij}^{box} + \mathbf{E}_i^{img} + \mathbf{E}_j^{reg}; i \le n, j \le m.$$

Then, they are fed into the visual encoder for better modeling the intramodal dynamics and enhancing the vision-specific order information.

$$\mathbf{S}_V^\ell = \text{SelfAttn}(\mathbf{H}_V^{\ell-1}) + \mathbf{H}_V^{\ell-1}, \ \mathbf{S}_V^\ell \in \mathbb{R}^{|\mathcal{V}| \times d_v},$$
$$\mathbf{H}_V^\ell = \text{FFN}(\mathbf{S}_V^\ell) + \mathbf{S}_V^\ell, \ \mathbf{H}_V^\ell \in \mathbb{R}^{|\mathcal{V}| \times d_v},$$

where $\mathbf{H}_V^{\ell-1}$ and $\mathbf{H}_V^\ell$ denote the inputs and outputs of the $\ell$-th encoder layer, respectively, and $\mathbf{H}_V^0$ is initialized as the $\mathbf{Z} = \{\mathbf{v}_{ij}\}_{i=1,j=1}^{i \le n, j \le m}$, and $d_v$ is the hidden dimension.

**Text-Vision Fusion.** Following Yu et al. (2021), the visual features are firstly injected by cross-modal multi-head attention (CrossMAttn):

$$\mathbf{M} = \text{CrossMAttn}(\mathbf{Q}, \mathbf{K}, \mathbf{V}), \ \mathbf{M} \in \mathbb{R}^{M \times d_c},$$

where $\mathbf{Q}$ are the projected textual features $\mathbf{Q} = \mathbf{H}_T^{N_e}\mathbf{W}_q$, $\mathbf{K}$ and $\mathbf{V}$ are the projected visual features with different weights, *i.e.*, $\mathbf{K} = \mathbf{H}_V^{N_v}\mathbf{W}_k$, $\mathbf{V} = \mathbf{H}_V^{N_v}\mathbf{W}_v$, and $\mathbf{Q} \in \mathbb{R}^{M \times d_c}$, $\mathbf{K}, \mathbf{V} \in \mathbb{R}^{|\mathcal{V}| \times d_c}$, and $d_c$ is the common hidden dimension.

Secondly, a forget gate $\mathbf{G}$ is used to filter redundant and noisy information from the visual features:

$$\mathbf{G} = \text{Sigmoid}(\text{Concat}(\mathbf{H}_T^{N_e}, \mathbf{M})\mathbf{W}_g + \mathbf{b}_g),$$
$$\mathbf{Z}_V = \mathbf{G} \otimes \mathbf{M}.$$

Finally, the vision-guided output $\mathbf{Z}_{T+V}$ is concatenated by $\mathbf{Z}_V$ and textual features $\mathbf{H}_T^{N_e}$, and then linearly project it to the original dimension $d$:

$$\mathbf{Z}_{T+V} = \text{Concat}(\mathbf{H}_T^{N_e}, \mathbf{Z}_V)\mathbf{W}_z + \mathbf{b}_z,$$

where Concat is the concatenation operation and $\mathbf{W}_*$ and $\mathbf{b}_*$ are trainable weights.

**Decoder.** The decoder follows a similar architecture but each of $N_d$ decoder layers has an additional multi-head cross-attention (CrossAttn) sub-layer:

$$\mathbf{S}_{dec}^\ell = \text{SelfAttn}(\mathbf{H}_{dec}^{\ell-1}) + \mathbf{H}_{dec}^{\ell-1}, \ \mathbf{S}_{dec}^{\ell-1} \in \mathbb{R}^{N \times d},$$
$$\mathbf{C}_{dec}^\ell = \text{CrossAttn}(\mathbf{S}_{dec}^\ell, \mathbf{Z}_{T+V}) + \mathbf{S}_{dec}^\ell, \qquad (1)$$
$$\mathbf{H}_{dec}^\ell = \text{FFN}(\mathbf{C}_{dec}^\ell) + \mathbf{C}_{dec}^\ell, \ \mathbf{C}_{dec}^\ell \in \mathbb{R}^{N \times d},$$

where $\mathbf{H}_{dec}^\ell \in \mathbb{R}^{N \times d}$ denotes the state of the $\ell$-th decoder layer. Then, at each decoding time step $t$, the top-layer ($N_d$-th) decoder hidden state $\mathbf{Z}_{dec,t}^{N_d}$ is fed into the softmax layer to produce the probability distribution of the next target token as:

$$p(y_t|\mathcal{X}_{L_1}, \mathcal{O}, y_{<t}) = \text{Softmax}(\mathbf{W}_o\mathbf{Z}_{dec,t}^{N_d} + \mathbf{b}_o),$$

where $\mathbf{W}_o$ and $\mathbf{b}_o$ are trainable weights.

Finally, the loss function is written as:

$$\mathcal{L}_{\text{MMS}}^{L_1,L_1} = -\sum_{t=1}^N \log(p(y_t^{L_1}|y_{<t}^{L_1}, \mathcal{X}^{L_1}, \mathcal{V})). \quad (2)$$

## 3 D²TV Training Framework

Based on the MMS model described in § 2.2, we firstly introduce the proposed *dual knowledge distillation* (DKD) method in § 3.1, which improves both MMS and MXLS tasks. Further, we present a simple yet effective *target-oriented contrastive objective* to filter needless visual information in § 3.2. Finally, we describe the *training and inference* in § 3.3.

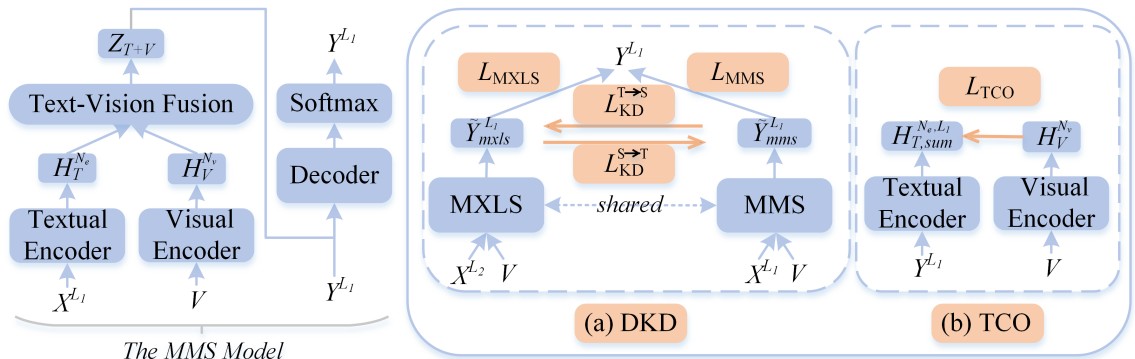

Figure 2: The overview of our model architecture. The left part is a general MMS model, which is enhanced by DKD and TCO. As shown in the right part, the (a) *dual knowledge distillation* (DKD) and (b) *target-oriented contrastive objective* (TCO), are proposed to improve the M³S model performance.

## 3.1 Dual Knowledge Distillation

As shown in the right part of Figure 2 (a), our framework involves training both MXLS and MMS models. Essentially, the MXLS model needs to simultaneously conduct machine translation and summarization (Liang et al., 2022d; Wang et al., 2022a,b) while the MMS model only conducts summarization. Obviously, it is harder to train an MXLS model than to learn an MMS model and that is why researchers (Nguyen and Luu, 2022; Liu et al., 2022) take the MMS model as the teacher to help the MXLS student model (*i.e., teacher→student* distillation). However, when the MXLS model achieves a level of multilingual and cross-lingual ability, the MXLS model can better transfer and share task knowledge among different languages. Therefore, the MXLS model, in turn, can guide the MMS model to conduct summarization in diverse languages (*e.g.,* English→English, Indonesian→Indonesian, Russian→Russian, and Urdu→Urdu), especially for low-resource ones (*i.e., student→teacher* distillation). That is why we propose DKD to mutually enhance their performance.

**Teacher→Student.** Specifically, for training the student model, given an input $\mathcal{X}^{L_2} = \{x_1^{L_2}, x_2^{L_2}, \ldots, x_{M_1}^{L_2}\}$ in language $L_2$ and corresponding visual features $\mathcal{V}$, the student model is to generate the cross-lingual summary $\mathcal{Y}^{L_1} = \{y_1^{L_1}, y_2^{L_1}, \ldots, y_N^{L_1}\}$ where $L_2 \neq L_1$. Then, we train the student model with two objectives as follows:

$$\mathcal{L}_{\text{student}}^{L_2,L_1} = \mathcal{L}_{\text{MXLS}}^{L_2,L_1} + \alpha \mathcal{L}_{\text{KD}}^{\text{T}(L_1,L_1)\to\text{S}(L_2,L_1)}, \quad (3)$$

where $\mathcal{L}_{\text{MXLS}}^{L_2,L_1}$ denotes by maximizing the likelihood of the ground-truth tokens which takes the cross-entropy form:

$$\mathcal{L}_{\text{MXLS}}^{L_2,L_1} = -\sum_{t=1}^{N} \log(p(y_t^{L_1}|y_{<t}^{L_1}, \mathcal{X}^{L_2}, \mathcal{V}), \quad (4)$$

and $\alpha$ is the trade-off factor and $\mathcal{L}_{\text{KD}}^{\text{T}(L_1,L_1)\to\text{S}(L_2,L_1)}$ represents KD loss to penalize the large distance of two hidden states of two summaries generated by the student and teacher models:

$$\mathcal{L}_{\text{KD}}^{\text{T}(L_1,L_1)\to\text{S}(L_2,L_1)} = \text{dist}(\mathbf{H}_{dec}^{N_d,\text{T}(L_1,L_1)}, \mathbf{H}_{dec}^{N_d,\text{S}(L_2,L_1)}), \quad (5)$$

where $\text{dist}(\cdot, \cdot)$ is the distance function to evaluate the difference between two representations (*e.g.,* KL and cosine similarity), and $\mathbf{H}_{dec}^{N_d,\text{T}(L_1,L_1)} = \{\mathbf{h}_1^T, \mathbf{h}_2^T, \ldots, \mathbf{h}_N^T\}$ denote the contextualized representations produced by the decoder of the teacher model, and $\mathbf{H}_{dec}^{N_d,\text{S}(L_2,L_1)} = \{\mathbf{h}_1^S, \mathbf{h}_2^S, \ldots, \mathbf{h}_N^S\}$ denote the representations from the decoder of the student model.

**Student→Teacher.** In particular, given the input document $\mathcal{X}^{L_1} = \{x_1^{L_1}, x_2^{L_1}, \ldots, x_{M_2}^{L_1}\}$ in language $L_1$ and corresponding visual features $\mathcal{V}$, the teacher model aims to generate its summary $\mathcal{Y}^{L_1}$ in the same language. We update the parameters of the teacher model with the following objective:

$$\mathcal{L}_{\text{teacher}}^{L_1,L_1} = \mathcal{L}_{\text{MMS}}^{L_1,L_1} + (1-\alpha)\mathcal{L}_{\text{KD}}^{\text{S}(L_2,L_1)\to\text{T}(L_1,L_1)}, \quad (6)$$

where $\mathcal{L}_{\text{KD}}^{\text{S}(L_2,L_1)\to\text{T}(L_1,L_1)}$ denotes the inverse KD loss:

$$\mathcal{L}_{\text{KD}}^{\text{S}(L_2,L_1)\to\text{T}(L_1,L_1)} = \text{dist}(\mathbf{H}_{dec}^{N_d,\text{S}(L_2,L_1)}, \mathbf{H}_{dec}^{N_d,\text{T}(L_1,L_1)}). \quad (7)$$

Finally, to flexibly distill the knowledge in Eq. 3 and Eq. 6, we apply an annealing strategy to dynamically adjust the balancing factor $\alpha$:

$$\alpha = \max(0.5, 1 - t1/T1), \quad (8)$$

where $t1$ is the training step ranging from 0 to the max training step $T$ and $T1$ is a hyperparameter. In this manner, the teacher model dominantly guides the student model in the first $T1/2$ training steps and the student model gradually distills the knowledge to the teacher. After training step $T1$, both models begin to equally distill their knowledge to each other.

## 3.2 Target-oriented Contrastive Objective

The $M^3S$ task requires a model to have the ability to understand and generate in multiple languages. However, there are some languages that are low-resource and lack enough data to train a good summarizer. Therefore, we aim to take visual features as the bridge between languages and hope that the visual features can be summary-oriented *i.e.*, discarding the noise that not appeared in the summary. To this end, we elaborately design an explicit target-oriented contrastive objective. Particularly, we push the visual feature $\mathcal{V}_i$ close to its corresponding summary $\mathcal{Y}_i^{L_1}$ and push apart irrelevant pairs, *e.g.*, $(\mathcal{V}_i, \mathcal{Y}_j^{L_1})$ where $i \neq j$. Therefore, we treat the paired $(\mathcal{V}_i, \mathcal{Y}_i^{L_1})$ as the positive sample and treat the pair $(\mathcal{V}_i, \mathcal{Y}_j^{L_1})$ as the negative samples where $i \neq j$. To obtain the representation of summary and image sequence, we apply *mean-pooling* with mask operation over the summary output $\mathbf{H}_{T,sum}^{N_e,L_1}$ of the $N_e$-th encoder layer and visual output $\mathbf{H}_V^{N_v}$ of the $N_v$-th encoder layer, respectively. That is, $\mathbf{h}_{sum}^{L_1} = \frac{1}{N} \sum_{k=1}^{N} (\mathbf{M}_k^{sum} \mathbf{h}_{sum,k}^{N_e,L_1})$, $\mathbf{h}_{sum}^{L_1} \in \mathbb{R}^d$, where $\mathbf{M}^{sum} \in \mathbb{R}^N$ denotes the mask matrix, whose value is either 1 or 0 indicating whether the token is padded. Similarly, we obtain the representation of image sequence, *i.e.*, $\mathbf{h}^{vis} = \frac{1}{m*n} \sum_{i=1}^{n} \sum_{j=1}^{m} (\mathbf{M}_{i,j}^{vis} \text{MLP}(\mathbf{h}_{i,j}^{N_v}))$, $\mathbf{h}^{vis} \in \mathbb{R}^d$, where $\mathbf{M}^{vis} \in \mathbb{R}^{n \times m}$ denotes the mask matrix and MLP is a fully-connected layer. Finally, the target-oriented contrastive training objective is defined by ($B$ is mini-batch size):

$$\mathcal{L}_{\text{TCO}}^{L_1} = -\log \frac{e^{\text{sim}(\mathbf{h}_i^{vis}, \mathbf{h}_{sum,i}^{L_1})/\tau}}{\sum_{b=1}^{B} e^{\text{sim}(\mathbf{h}_i^{vis}, \mathbf{h}_{sum,b}^{L_1})/\tau}}, \quad (9)$$

where $\text{sim}(\cdot, \cdot)$ is the cosine similarity and $\tau$ denotes a temperature hyperparameter.

## 3.3 Training and Inference

At training, we train our model with the following objective:

$$\mathcal{J} = \sum_{i=1}^{K} \sum_{j=1, j \neq i}^{K} (\mathcal{L}_{\text{student}}^{L_j, L_i} + \mathcal{L}_{\text{teacher}}^{L_i, L_i} + \beta \mathcal{L}_{\text{TCO}}^{L_i}), \quad (10)$$

where $K$ is the number of languages and $\beta$ is balancing hyper-parameter.

Note that the MMS model and the MXLS model are shared and thus the final model can conduct summarization in any language. During inference, the training objectives are not involved and only the model is used to generate summaries.

# 4 Experiments

## 4.1 M³Sum Dataset

There is no many-to-many multimodal summarization benchmark dataset until now. We construct one as follows. Based on the CrossSum dataset (Bhattacharjee et al., 2022) and MM-Sum dataset (Liang et al., 2022b), we construct a **M**any-to-**M**any **M**ultimodal **Sum**marization (M³Sum) dataset. The original CrossSum dataset is crawled from the BBC website[3] and its quality has been verified and ensured reliability by Bhattacharjee et al. (2022). However, the lack of associated image sequence in CrossSum, makes it impossible to directly conduct research on MMS and MXLS. The original MM-Sum dataset is also crawled from the BBC website, which includes multilingual multimodal summarization. But it cannot conduct cross-lingual summarization due to the lacking of cross-lingual alignment. Therefore, we reorganize both datasets and conduct cross-lingual alignment through the same *url* in each dataset.

According to the dataset size of each language, we follow CrossSum (Bhattacharjee et al., 2022) and utilize about 80% training:10% validation:10% test splitting. Besides, in CrossSum, the number of languages is 44 and thus there are 44*44 language directions. Tab. 4 of Appendix A shows the detailed statistic of our M³Sum and please refer to it for details.

## 4.2 Setup and Metrics

**Implementation Details.** For efficiency, we randomly select 4 languages (*i.e.*, English, Indonesian, Russian, and Urdu), which totally cover 16 language directions. Please refer to Appendix B for

---

[3] https://www.bbc.com/

| Src \ Trg | Models | English | Indonesian | Russian | Urdu | Avg. |
|---|---|---|---|---|---|---|
| **English →** | MMS | **36.16 / 13.08 / 27.67 / 70.57** | 6.87 / 1.94 / 6.34 / 63.39 | 1.23 / 0.20 / 1.21 / 59.60 | 0.14 / 0.00 / 0.14 / 55.44 | 11.09 / 3.80 / 8.84 / 62.25 |
| | MXLS | 6.94 / 2.35 / 6.03 / 61.82 | 27.23 / 9.32 / 22.13 / 68.40 | 22.52 / 7.88 / 18.07 / 64.84 | 32.27 / 11.17 / 25.15 / 68.29 | 22.24 / 7.68 / 17.84 / 65.83 |
| | MMS+MXLS | 35.80 / 13.45 / 27.93 / 70.64 | 27.18 / 9.20 / 22.04 / 68.78 | 23.88 / 8.03 / 19.30 / 65.57 | 28.59 / 8.94 / 22.95 / 66.79 | 28.86 / 9.90 / 23.07 / 67.94 |
| | Vanilla-KD | 34.60 / 12.70 / 26.86 / 70.07 | 27.75 / 9.71 / 22.63 / 68.93 | 24.36 / 8.00 / 19.41 / 65.42 | 31.53 / 10.28 / 24.83 / 67.76 | 29.56 / 10.17 / 23.43 / 68.04 |
| | D²TV (Ours) | 36.12 / 13.21 / **27.99 / 70.64** | **28.87 / 10.26 / 23.77 / 69.31** | **25.53 / 8.69 / 20.72 / 66.01** | **32.56 / 10.73 / 25.71 / 68.39** | **30.77 / 10.72 / 24.53 / 68.83** |
| **Indonesian →** | MMS | 7.28 / 2.03 / 6.73 / 63.59 | 34.10 / 13.92 / 27.92 / 71.14 | 1.19 / 0.19 / 1.16 / 60.40 | 0.13 / 0.01 / 0.13 / 56.42 | 10.67 / 4.03 / 8.98 / 62.88 |
| | MXLS | 32.43 / 11.31 / 25.09 / 69.13 | 5.39 / 1.59 / 4.88 / 61.82 | 21.65 / 7.80 / 17.62 / 65.09 | 31.85 / 11.00 / 25.39 / 68.53 | 22.83 / 7.92 / 18.24 / 66.14 |
| | MMS+MXLS | 32.59 / 11.67 / 25.42 / 69.53 | **34.43 / 14.56 / 28.43 / 71.30** | 24.38 / 8.70 / 20.01 / 66.18 | 30.65 / 10.30 / 24.95 / 67.90 | 30.51 / 11.31 / 24.70 / 68.72 |
| | Vanilla-KD | 32.88 / 11.56 / 25.45 / 69.46 | 32.67 / 13.01 / 26.71 / 70.68 | 25.50 / 8.97 / 20.65 / 66.30 | 32.48 / 11.31 / 25.88 / 68.79 | 30.88 / 11.21 / 24.67 / 68.80 |
| | D²TV (Ours) | **34.54 / 12.10 / 26.50 / 69.73** | 33.94 / 14.08 / 28.05 / 71.19 | **26.40 / 9.27 / 21.35 / 66.56** | **33.45 / 11.38 / 26.60 / 68.89** | **32.08 / 11.71 / 25.63 / 69.09** |
| **Russian →** | MMS | 1.24 / 0.22 / 1.23 / 60.20 | 1.23 / 0.22 / 1.19 / 60.86 | **30.30 / 11.82 / 24.25 / 68.16** | 0.11 / 0.00 / 0.11 / 56.10 | 8.22 / 3.02 / 6.69 / 61.33 |
| | MXLS | 29.47 / 9.86 / 22.82 / 68.10 | 25.78 / 9.06 / 21.01 / 68.20 | 2.68 / 0.90 / 2.41 / 59.02 | 31.06 / 10.64 / 25.05 / 68.08 | 22.24 / 7.61 / 17.82 / 65.85 |
| | MMS+MXLS | 30.79 / 9.82 / 24.02 / 68.74 | 27.37 / 9.84 / 22.70 / 69.13 | 29.67 / 11.67 / 24.33 / 68.04 | 30.53 / 10.04 / 24.92 / 67.95 | 29.59 / 10.37 / 24.09 / 68.50 |
| | Vanilla-KD | 31.18 / 10.64 / 24.26 / 68.70 | 26.50 / 9.60 / 21.91 / 68.73 | 28.32 / 10.93 / 23.00 / 67.37 | 31.38 / 10.76 / 25.25 / 68.33 | 29.34 / 10.48 / 23.60 / 68.28 |
| | D²TV (Ours) | **32.87 / 11.06 / 25.59 / 69.14** | **29.03 / 10.59 / 23.64 / 69.59** | 29.90 / 11.44 / **24.75 / 68.20** | **33.29 / 11.88 / 26.96 / 69.10** | **31.27 / 11.24 / 25.13 / 68.96** |
| **Urdu →** | MMS | 0.09 / 0.00 / 0.09 / 55.58 | 0.05 / 0.00 / 0.05 / 56.43 | 0.09 / 0.00 / 0.09 / 56.03 | 37.54 / 15.04 / 30.19 / 70.55 | 9.44 / 3.76 / 7.60 / 59.64 |
| | MXLS | 29.95 / 9.06 / 23.09 / 68.36 | 26.00 / 9.37 / 21.44 / 68.43 | 21.52 / 6.87 / 17.23 / 65.35 | 7.70 / 2.62 / 6.15 / 58.76 | 21.29 / 6.98 / 16.97 / 65.22 |
| | MMS+MXLS | 28.94 / 9.29 / 22.81 / 67.76 | 26.43 / 8.84 / 21.74 / 68.55 | 20.47 / 6.44 / 16.74 / 64.33 | 37.72 / 15.78 / 30.97 / 70.96 | 28.39 / 10.17 / 23.14 / 67.90 |
| | Vanilla-KD | 29.60 / 9.63 / 23.00 / 67.88 | 26.30 / 9.02 / 21.65 / 68.29 | 22.58 / 7.36 / 18.03 / 65.22 | 37.52 / 15.25 / 30.19 / 70.56 | 29.00 / 10.31 / 23.21 / 67.98 |
| | D²TV (Ours) | **32.01 / 9.99 / 24.71 / 68.65** | **28.23 / 10.01 / 23.19 / 69.25** | **24.52 / 7.87 / 19.98 / 66.13** | **38.05 / 16.12 / 31.30 / 70.97** | **30.70 / 10.91 / 24.71 / 68.75** |

(a) The test results based on the mT5 backbone in terms of ROUGE-1 / ROUGE-2 / ROUGE-L / BERTSCORE scores.

| Src \ Trg | Models | English | Indonesian | Russian | Urdu | Avg. |
|---|---|---|---|---|---|---|
| **English →** | MMS | 34.19 / 11.87 / 26.14 / 69.38 | 24.70 / 6.94 / 19.55 / 67.37 | 21.50 / 6.14 / 17.14 / 64.08 | 23.18 / 5.00 / 17.68 / 63.04 | 25.89 / 7.49 / 20.13 / 65.96 |
| | MXLS | 26.06 / 8.49 / 20.22 / 64.50 | 25.89 / **8.41** / 21.03 / 66.96 | **24.87** / 7.96 / 20.01 / 65.32 | 27.30 / 7.76 / 21.31 / 65.65 | 25.27 / 8.06 / 20.08 / 65.58 |
| | MMS+MXLS | **35.03 / 12.50 / 27.17 / 69.75** | 22.97 / 7.37 / 18.65 / 68.05 | 23.50 / 7.81 / 18.95 / 65.72 | 27.14 / 8.04 / 21.17 / 66.51 | 27.16 / 8.93 / 21.49 / 67.45 |
| | Vanilla-KD | 34.84 / **12.52** / 26.98 / 69.46 | 24.29 / 7.76 / 19.81 / 67.98 | 24.49 / 7.80 / 19.64 / 65.76 | **29.06 / 8.83 / 22.84 / 66.90** | 28.17 / 9.22 / 22.31 / 67.47 |
| | D²TV (Ours) | 34.78 / 12.36 / 26.81 / 69.55 | **26.13 / 8.39 / 21.15 / 68.26** | 24.84 / **8.28 / 20.06 / 65.94** | 28.60 / 8.44 / 22.30 / 66.70 | **28.59 / 9.37 / 22.58 / 67.71** |
| **Indonesian →** | MMS | 30.79 / 9.09 / 23.03 / 67.68 | 30.12 / 10.77 / 24.16 / 69.14 | 21.67 / 6.27 / 17.46 / 64.28 | 25.13 / 5.27 / 19.01 / 63.82 | 26.93 / 7.85 / 20.92 / 66.23 |
| | MXLS | 32.85 / 10.56 / 24.78 / 68.53 | 18.27 / 6.45 / 15.00 / 63.64 | 22.58 / 7.29 / 18.07 / 63.65 | 25.96 / 8.10 / 20.70 / 63.36 | 24.50 / 8.06 / 19.36 / 64.99 |
| | MMS+MXLS | 33.87 / 11.51 / 26.14 / 69.36 | 29.81 / 11.33 / 24.29 / 69.75 | 24.26 / 8.40 / 19.65 / 65.89 | 29.05 / **8.99** / 22.87 / 66.88 | 29.25 / 10.06 / 23.24 / 67.97 |
| | Vanilla-KD | 33.68 / 11.68 / 25.80 / 69.25 | 30.49 / 11.58 / 24.84 / 69.59 | 24.22 / 8.28 / 19.48 / 66.05 | 29.16 / 8.77 / 23.05 / 67.05 | 29.38 / 10.07 / 23.29 / 67.98 |
| | D²TV (Ours) | **34.18 / 11.75 / 26.17 / 69.46** | **31.25 / 11.75 / 25.30 / 69.93** | **24.99 / 8.66 / 20.29 / 66.38** | **29.56** / 8.88 / **23.18 / 67.28** | **30.00 / 10.26 / 23.74 / 68.26** |
| **Russian →** | MMS | 29.53 / 8.39 / 22.53 / 66.86 | 24.02 / 6.73 / 19.21 / 66.84 | 28.29 / 9.85 / 22.33 / 66.88 | 24.79 / 5.16 / 18.92 / 63.59 | 26.66 / 7.53 / 20.75 / 66.04 |
| | MXLS | 32.01 / 10.83 / 24.39 / 67.87 | 23.92 / 8.20 / 19.36 / 65.78 | 23.41 / 8.19 / 18.70 / 62.44 | 24.59 / 7.54 / 19.39 / 63.58 | 25.23 / 8.47 / 19.93 / 64.91 |
| | MMS+MXLS | 32.94 / 11.40 / 25.74 / 68.79 | 24.58 / 8.43 / 20.09 / 67.92 | 28.10 / 10.37 / 22.66 / 66.93 | 27.44 / 8.54 / 21.59 / 66.52 | 28.27 / 9.68 / 22.52 / 67.54 |
| | Vanilla-KD | 32.86 / 11.54 / 25.64 / 68.75 | 24.63 / 8.40 / 20.15 / 68.12 | 28.27 / 10.31 / 22.64 / 67.04 | **28.50** / 8.87 / 22.56 / 66.83 | 28.56 / 9.78 / 22.75 / 67.69 |
| | D²TV (Ours) | **33.61 / 11.65 / 26.09 / 69.14** | **26.57 / 8.96 / 21.63 / 68.47** | **28.39 / 10.47 / 22.92 / 67.38** | 28.39 / 8.81 / **22.68 / 66.88** | **29.24 / 9.97 / 23.33 / 67.96** |
| **Urdu →** | MMS | 24.55 / 5.14 / 18.29 / 64.04 | 19.20 / 4.39 / 15.06 / 64.68 | 17.22 / 3.59 / 13.36 / 61.99 | 34.82 / 12.63 / 27.14 / 68.99 | 23.95 / 6.44 / 18.46 / 64.92 |
| | MXLS | 30.89 / 9.53 / 23.82 / 67.05 | 22.74 / 7.40 / 18.59 / 65.59 | 21.63 / 6.62 / 17.66 / 63.27 | 21.90 / 6.94 / 17.19 / 60.13 | 23.69 / 7.49 / 18.84 / 64.01 |
| | MMS+MXLS | 31.54 / 10.42 / 24.53 / 68.24 | 22.94 / 7.62 / 18.88 / 67.02 | 22.32 / 7.13 / 18.32 / 64.93 | 35.86 / 13.49 / 28.48 / 69.56 | 28.17 / 9.66 / 22.55 / 67.43 |
| | Vanilla-KD | 30.96 / 10.20 / 24.34 / 68.18 | 23.72 / 8.16 / 19.49 / 67.36 | 21.94 / 6.79 / 17.90 / 64.69 | 35.83 / 13.53 / 28.39 / 69.50 | 28.11 / 9.67 / 22.53 / 67.43 |
| | D²TV (Ours) | **31.65 / 10.63 / 24.90 / 68.62** | **25.47 / 8.52 / 20.68 / 67.75** | **22.38 / 7.19 / 18.44 / 65.37** | **36.46 / 13.76 / 28.75 / 69.94** | **28.99 / 10.03 / 23.19 / 67.92** |

(b) The test results based on the mBART-50 backbone.

Table 1: The block in " *\*/\*/\*/\** " denotes the MMS results and the block in " *\*/\*/\*/\** " indicates the MXLS results. The " *\*/\*/\*/\** " indicates the average (Avg.) score for each model and the best scores in each block are **bold**. Our **bold** results indicate that statistically significantly better than the "Vanilla-KD" with t-test $p < 0.05$. Note that the results out of each block (e.g., English→English block) cannot be compared to others (e.g., Indonesia→English block) because they belong to different language directions. Therefore, in each block of MMS, the MMS always surpasses MXLS without any exception. In each block of MXLS, the MXLS always surpasses MMS without any exception.

more implementation details including data pre-processing and hyper-parameters settings.

**Metrics.** Following Bhattacharjee et al. (2022); Wang et al. (2023a), we use the standard ROUGE scores (ROUGE-1, ROUGE-2, and ROUGE-L) (Lin, 2004) with the statistical significance test (Koehn, 2004) for a fair comparison. Besides, we apply BERTSCORE (Zhang et al., 2020) for a comprehensive comparison.

### 4.3 Comparison Models

- **MMS**: It is the MMS model trained with the objective Eq. 2.
- **MXLS**: It is the MXLS model trained with the objective Eq. 4.
- **MMS+MXLS**: It is the model jointly trained with the objectives Eq. 2 and Eq. 4, which actu-

ally is the M³S training objective.
- **Vanilla-KD**: It is the model enhanced with the knowledge distillation, which is trained with the objectives Eq. 2, Eq. 4 and Eq. 3.
- **D²TV** : It is the proposed model which are trained with the objective Eq. 10.

All the above models use the multimodal Transformer described in § 2.2 and involve two strong training backbones: **mT5** (Xue et al., 2021) and **mBART-50** (Tang et al., 2021).

### 4.4 Main Results

Tab. 1 presents the main results on many-to-many scenarios grounding on different backbones. Overall, our model obtains significantly better results than all contrast models in both settings.

**Results based on mT5 backbone.** In Tab. 1 (a),

| | Models | English→* | Indonesian→* | Russian→* | Urdu→* | Train (S) |
|---|---|---|---|---|---|---|
| 0 | MMS+MXLS | 28.86 / 9.90 / 23.07 / 67.94 | 30.51 / 11.31 / 24.70 / 68.72 | 29.59 / 10.37 / 24.09 / 68.50 | 28.39 / 10.17 / 23.14 / 67.90 | 6.12 |
| 1 | *w/o* Visual Features | 28.48 / 9.44 / 22.73 / 67.71 | 30.12 / 10.83 / 24.33 / 68.39 | 29.18 / 10.01 / 23.68 / 68.33 | 27.91 / 9.88 / 22.79 / 67.67 | 5.37 |
| 2 | *w/* Vanilla KD | 29.56 / 10.17 / 23.43 / 68.04 | 30.88 / 11.21 / 24.67 / 68.80 | 29.34 / 10.48 / 23.60 / 68.28 | 29.00 / 10.31 / 23.21 / 67.98 | 7.95 |
| 3 | *w/* DKD | 30.19 / 10.63 / 23.98 / 68.48 | 31.49 / 11.44 / 25.12 / 68.92 | 30.89 / 10.88 / 24.51 / 68.55 | 29.96 / 10.65 / 24.38 / 68.51 | 9.58 |
| 4 | *w/* CAT | 30.19 / 10.57 / 24.03 / 68.45 | 31.42 / 11.49 / 25.33 / 68.88 | 30.75 / 10.59 / 24.41 / 68.64 | 29.96 / 10.61 / 24.32 / 68.31 | 13.45 |
| 5 | *w/* TCO | 30.01 / 10.45 / 23.77 / 68.27 | 31.27 / 11.25 / 25.01 / 68.85 | 30.78 / 10.61 / 24.45 / 68.51 | 29.89 / 10.44 / 24.15 / 68.29 | 8.90 |
| 6 | *w/* DKD&TCO | **30.77 / 10.72 / 24.53 / 68.83** | **32.08 / 11.71 / 25.63 / 69.09** | **31.27 / 11.24 / 25.13 / 68.96** | **30.70 / 10.91 / 24.71 / 68.75** | 11.88 |

Table 2: Ablation study based on the mT5 ( Avg. results of ROUGE-1 / ROUGE-2 / ROUGE-L / BERTSCORE), where each component is separately added on the "MMS+MXLS". "*" denotes the four languages (*i.e.*, English, Indonesian, Russian, and Urdu). The "CAT" denotes the complex auxiliary tasks of (Liang et al., 2022b). Train (S) denotes how many seconds are required for each model to train one step (32 batch size * 8 GPUs).

1) in each group (*e.g.*, English→{English, Indonesian, Russian, Urdu}), the MMS model typically performs better in generating monolingual summaries while it cannot process well in cross-lingual settings. The reason is that the MMS model has no access to the cross-lingual data during training. The MXLS model faces a similar phenomenon where it cannot handle well the monolingual summaries while generating better cross-lingual summaries. In contrast, the "MMS+MXLS" model, as a multitask model, achieves better results than both MMS and MXLS models, showing that the MMS and MXLS tasks can benefit each other and thus improve both of them. Based on this finding, a dual knowledge distillation is more reasonable than unidirectional knowledge distillation. Our results further demonstrate this (See ablation study). 2) Generally, in each block, we find that our D$^2$TV approach notably outperforms the Vanilla-KD method, showing the effectiveness of dual knowledge distillation and target-oriented contrastive learning. Although our results are slightly worse than the MMS model in English→English, Indonesian→Indonesian, and Russian→Russian directions of " */*/*/* " blocks, our D$^2$TV model can balance well between MMS and MXLS. The results in each " Avg. " blocks fully prove this point. 3) On average, our model consistently and significantly surpasses all baselines by large margins (*e.g.*, the previous best "Vanilla-KD", up to **1.70/0.60/1.50** ROUGE and **0.77** BERTScore scores in Urdu→* directions, respectively).

**Results based on mBART-50 backbone.** In Tab. 1 (b), we observe similar findings as in the mT5-based scenario. This demonstrates that our conclusions are solid and convincing on general pre-trained language models. All these results prove the superiority of our approach.

## 5 Analysis

### 5.1 Ablation Study

We conduct ablation studies to investigate how well each component works. The results are shown in Tab. 2. We have the following conclusions:

- (Row 1 vs. row 0). The results show that incorporating visual features has a positive impact on the model performance, demonstrating the importance of image sequence for the summary.

- (Row 2 vs. row 0). The vanilla KD makes reasonable contributions, showing that the MMS model indeed helps improve the quality of summaries in terms of both ROUGE and BERTScore scores, suggesting that distilling the knowledge of MMS to MXLS is helpful to summarization;

- (Row 3 vs. row 2&row 0). The results show that dual knowledge distillation further improves the model performance, indicating that the knowledge of MMS and MXLS are beneficial to each other and thus can enhance both of them.

- (Row 5 vs. row 4&row 0). The results show that summary-oriented visual features can significantly improve the quality of summaries and our simple TCO achieves comparable performance with the CAT with less training time. This shows the superiority of the target-oriented contrastive objective.

- (Row 6 vs. row 0). Adding DKD and TCO exhibit notable cumulative benefits, showing the effectiveness of the proposed approach.

### 5.2 Human Evaluation

Following Liang et al. (2022b), we conduct human studies on 50 samples randomly selected from English→English and Russian→English test sets to further evaluate the performance of all models. We invite three Chinese postgraduate students who major in English to compare the generated sum-

| Models | English→English | | | Russian→English | | |
|---|---|---|---|---|---|---|
| | Flu. | Con. | Inf. | Flu. | Con. | Inf. |
| MMS | 3.28 | 3.04 | 2.58 | 1.20 | 1.04 | 0.88 |
| MXLS | 1.22 | 1.10 | 0.96 | 2.72 | 2.28 | 2.14 |
| MMS+MXLS | 3.44 | 3.28 | 3.18 | 3.26 | 3.16 | 3.04 |
| Vanilla-KD | 3.60 | 3.46 | 3.22 | 3.40 | 3.28 | 3.18 |
| D$^2$TV | **4.14** | **3.94** | **3.78** | **3.86** | **3.58** | **3.50** |

Table 3: Human evaluation results. 'Flu.': fluency, 'Con.': conciseness, and 'Inf.': informativeness.

maries [4] and assess each summary from three independent aspects: **fluency** (Flu.), **conciseness** (Con.) and **informativeness** (Inf.). We ask them to score each aspect from 1 (worst) to 5 (best). The average results are presented in Tab. 3.

Tab. 3 shows the human results. We find that our D$^2$TV substantially outperforms all contrast models under all criteria in both directions, which further shows the effectiveness and superiority of our approach. The Fleiss' Kappa scores (Fleiss and Cohen, 1973) of Flu., Con. and Inf. are 0.74, 0.70 and 0.65, respectively, which indicates a substantial agreement among three evaluators. Furthermore, we present a case study in Appendix C and it intuitively shows the superiority of our D$^2$TV.

# 6 Related Work

**Multimodal Monolingual Summarization (MMS).** With the rapid growth of multimedia, many MMS datasets have been built which cover video summarization (Tjondronegoro et al., 2011; Sanabria et al., 2018), movie summarization (Evangelopoulos et al., 2013), meeting records summarization (Erol et al., 2003), sentence summarization (Li et al., 2018a, 2017), product summarization (Li et al., 2020a), and news summarization (Zhu et al., 2018; Chen and Zhuge, 2018; Hasan et al., 2021; Fu et al., 2021; Liang et al., 2022b). With the data resources extensively used, the MMS task has attracted much attention, where the existing work mainly focuses on 1) how to effectively exploit the additional features which are generally implicitly learned by the MMS objective or 2) explicit and complex auxiliary tasks, having achieved impressive performance on these high-resource English datasets (Li et al., 2018b, 2020b; Zhu et al., 2020, 2021; Zhang et al., 2021b,a; Yu et al., 2021). In this work, we instead of focusing on introducing a more general and practical many-to-many multimoal summarization

---

[4]When evaluating summaries in Russian→English, we show them the English document rather than the Russian document where the English and Russian document describe the same thing.

setting and also provide a corresponding benchmark dataset. Additionally, we propose a simple yet effective target-oriented contrastive learning objective to filter needless visual features, *i.e.*, offer summary-oriented visual features.

**Multimodal Cross-lingual Summarization (MXLS).** There is only one study that focuses on the MXLS task, *i.e.*, Liu et al. (2022) first propose this task and design a triple-stage training framework and distill the knowledge from MMS to enhance MXLS while ignoring the performance of MMS. Different from this work, we introduce the many-to-many multimodal summarization task. Furthermore, we devise a dual knowledge distillation approach to simultaneously improve both MMS and MXLS tasks.

**Knowledge Distillation (KD).** KD (Hinton et al., 2015) is to transfer the knowledge (*e.g.*, soft targets outputs) of the stronger model (aka. the teacher model) to the small model (aka. the student model), which has achieved impressive results in the literature (Zhang et al., 2023). In summarization, (Zhang et al., 2021b) adopt KD from a vision-language pre-trained model to improve image selection when generating multimodal summaries. Besides, researchers (Nguyen and Luu, 2022; Liu et al., 2022) typically treat the monolingual summarization model as the teacher model and the cross-lingual one as the student model because the monolingual summarization model is easier to train well than the cross-lingual one, which has shown promising performance on cross-lingual summarization task while ignoring the performance of the monolingual one. In this work, we aim to mutually prompt both monolingual and cross-lingual summarization tasks via dual KD rather than only improving the cross-lingual summarization task by unidirectional KD.

**Constrastive Learning.** The idea of contrastive learning aims to learn effective representation by pulling semantically close neighbors together and pushing apart non-neighbors (Hadsell et al., 2006), which has verified its superiority in many fields (Zhou et al., 2023). In summarization, Liu and Liu (2021) use contrastive loss to post-rank generated summaries and achieves good results in textual-only benchmark datasets. Cao and Wang (2021) and Xu et al. (2021) use contrastive learning to improve faithfulness and factuality and observe consistent improvements. Wang et al. (2021) apply contrastive learning for multilingual summarization

and obtain promising performance. Differently, we introduce it into the multimodal area and aim to pull the visual feature close to its corresponding summary and offer summary-oriented visual features. Therefore, we can improve the quality of summaries from the perspective of visual features rather than the textual document.

## 7 Conclusion

In this paper, we first introduce a more general task, *i.e.*, M$^3$S, which can support both MMS and MXLS tasks. Further, we propose a dual knowledge distillation and target-oriented vision (D$^2$TV) enhanced framework for the new task. Extensive experiments demonstrate that our model significantly outperforms related baselines in terms of ROUGE, BERTScore scores, and human evaluation. Furthermore, we contribute a many-to-many multimodal summarization (M$^3$Sum) dataset to the research community.

## Limitations

Although we show that our D$^2$TV outperforms the vanilla-kD model based on two stronger backbone *i.e.*, mT5 (Xue et al., 2021) and mBART-50 (Tang et al., 2021), there are some limitations worth considering to study in future work: (1) In this study, we only provide 44 languages and conduct experiments on four out of them, and future work could extend our method to more languages; (2) With the development of the large-scale language models, extending and validating our approach on them may be future work.

## Ethics Statement

In this section, we consider the potential ethical issues of our model. In this paper, we propose D$^2$TV which is trained on the publicly-available BBC datasets. Therefore, D$^2$TV might lead to incorrect summaries in applications and involve the same biases and toxic behaviors exhibited by the datasets. Besides, we obtained our M$^3$Sum dataset by reorganizing the CrossSum (Bhattacharjee et al., 2022) and MMSum (Liang et al., 2022b) datasets[5] and its permissions are granted to copy, distribute and modify the contents under the terms of the Creative Commons AttributionShareAlike 3.0 Unported License and Creative Commons CC0 License, respectively.

---

[5]The data originally comes from: https://www.bbc.com/

## Acknowledgements

The research work described in this paper has been supported by the National Key R&D Program of China (2020AAA0108001)and the National Nature Science Foundation of China (No. 61976015, 61976016, 61876198 and 61370130). The authors would like to thank the anonymous reviewers for their valuable comments and suggestions to improve this paper.

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

## A Dataset Statistics and Splits.

As shown in Tab. 4, we only present 4*4 language directions of our M³Sum used in this work for simplicity. Actually, our M³Sum covers 44*44 languages and in total includes 1,078,215 article-summary pairs with 3,479,348 images, where each article-summary pair contains about 3.23 images on average. The average article and summary length for all languages is about 520 and 84, respectively. According to the dataset size of each language, we follow CrossSum (Bhattacharjee et al., 2022) and utilize about 80% training:10% validation:10% test splitting. Besides, in CrossSum, the number of languages is 44 and thus there are 44*44 language directions. For efficiency, we randomly select 4 languages (*i.e.*, English, Indonesian, Russian, and Urdu), which totally cover 16 language directions.

| Languages | English | Indonesian | Russian | Urdu |
|---|---|---|---|---|
| **English** | 24,768 | 10,037 | 9,076 | 6,297 |
| **Indonesian** | 9,814 | 23,176 | 7,260 | 6,324 |
| **Russian** | 8,902 | 7,329 | 21,036 | 5,179 |
| **Urdu** | 6,052 | 5,810 | 4,700 | 17,800 |

Table 4: An example of 4 * 4 Language pairs covered by our M³Sum dataset.

## B Implementation Details

**Data Pre-Processing.** Following Bhattacharjee et al. (2022), we pre-process the textual data by truncating or padding them into sequences of 512 tokens for $\mathcal{X}$ and the outputs $\mathcal{Y}$ to 84 tokens after using the 250k wordpiece (Xue et al., 2021) vocabulary provided with the mT5 checkpoint (similar to mBART-50 setting). For the image sequence, following Liang et al. (2022b), we truncate or pad the sequence length to 180 (*i.e.*, five images: 5 * 36; n=5, m=36).

**Hyper-Parameters.** In this work, we use two strong backbones, *i.e.*, the $base$[6] model of mT5 (Xue et al., 2021) and the $large$[7] model of mBART-50 (Tang et al., 2021). We list detailed hyper-parameter used in this work in Tab. 5.

For inference, we use beam search with beam size 4 and length penalty of $\gamma = 0.6$. When calculating the ROUGE scores, we use the multi-lingual

---

| Hyperparameters | mT5 | mBART-50 |
|---|---|---|
| batch size ($B$) | 32 | 32 |
| number of GPUs | 8 V100 | 8 V100 |
| hidden size | 768 | 1024 |
| filter size | 2048 | 4096 |
| encoder layers | 12 | 12 |
| decoder layers | 12 | 12 |
| attention heads | 12 | 16 |
| label smoothing | 0.1 | 0.1 |
| learning rate | 2e-5 | 5e-6 |
| warmup steps | 2,000 | 2,000 |
| training steps $T$ | 10,000 | 10,000 |
| $T1$ | 5,000 | 5,000 |
| training time | ≈33h | ≈8h |
| optimizer | Adam | Adam |
| adam beta1 | 0.9 | 0.9 |
| adam beta2 | 0.998 | 0.98 |
| layer normalization | postnorm | postnorm |
| $M$ | 520 | 520 |
| $N$ | 84 | 84 |
| $m$ | 5 | 5 |
| $n$ | 36 | 36 |
| $N_e$ | 12 | 12 |
| $N_d$ | 12 | 12 |
| $N_v$ | 4 | 4 |
| $d$ | 768 | 1024 |
| $d_v$ | 2048 | 2048 |
| $d_c$ | 256 | 256 |
| $K$ | 4 | 4 |
| $\beta$ | 1.0 | 1.0 |

Table 5: Training hyperparameters and model configurations of our experiments.

rouge[8] toolkit following Bhattacharjee et al. (2022). All experimental results reported in this paper are the average of three runs with different random seeds.

## C Case Study

Fig. 3 shows an example of the many-to-many multimodal summarization, the generated summary, and the ground truth summary in different languages. (updating later.)

---

**Article:** China: The story of China has been one of extraordinary growth in the last decade, but there have been recent concerns that there will be a significant economic slowdown. One worry is that this would trigger panicked reactions from domestic investors and lead to a stock market crash. With China establishing its Shanghai stock exchange only in 1990, its market is considered immature compared to the rest of the world. The shares are almost entirely owned by domestic traders, many of whom are 'mom and pop' investors with little experience in investing. The lack of large, experienced and professional organisations as investors means that the market can be much more volatile.

Central bank: Over the last few months, China's central bank has been repeatedly propping up the stock market to ensure stability. They have been doing it through several big measures, such as cutting central bank interest rates - which allows more money to flow easily - and buying up shares to stop them from falling. After losses last week, there was an expectation on Friday that there would be yet another such drastic move. But that did not happen - causing panic to ripple out and a dramatic drop in shares on Monday. The stock market saw its worst single-day plunge since 2007.

Currency: One of the possible triggers for the drop in past trading sessions was the earlier decision by the central bank to devalue the yuan and allow it to trade more flexibly. Unlike most currencies, the Chinese currency is not allowed to trade freely according to the number of buyers and sellers in international markets. Rather, the central bank sets a daily rate to the US dollar and for the rest of the day, the yuan is allowed to trade 2% up or down from that rate. Earlier in August, the bank cut that rate by almost 2%, sending a first wave of insecurity through markets. The move was seen as an attempt to help exports by making Chinese goods cheaper abroad. The central bank also said it would set the daily rate based on how the yuan traded the previous day, which means that it could fall a lot further in future.

Contagion: China's stock market slump caused investor uncertainty to spread across the region and then around the globe, destabilising stock markets in New York and Europe. This knock-on effect has highlighted how much of a linchpin China's stock market is in the global marketplace. Hong Kong-based investment analyst Peter Churchouse says China's market was "irrelevant" 35 years ago and as recent as a decade ago, it merely followed trends in the global economy. But now the tables have turned, he says. "The global economy and global markets have a 'Made in China' label on them."

Correction or Crash? Monday's global turmoil sparked fears of another international financial meltdown, but analysts say it was merely over-inflated markets correcting themselves. They are however warning of further slumps in the long run. Nicholas Teo of financial analysis firm CMC Markets says financial markets were "intoxicated" by easy and cheap funding in recent years, boosting stocks' value and consumer spending. The turmoil caused by China's stock slump "suggested that the great unwind of the excesses is beckoning". As for China itself, analysts say that as the market matures over the years and investors become more experienced, it will become less volatile. This could also happen if China's government removes some of the restrictions that hinder foreign ownership of shares, thus paving the way for bigger more professional firms to come in and inject stability. Currently foreigners only own 2% of stocks.

Consequence: Observers have described this incident as a "rude awakening" for global investors who have paid scant attention to China. Extreme movements in China's market will probably become a more common sight, given its peculiarity of being dominated by small-time inexperienced investors. Meanwhile China's economy is still expected to slow which in turn would affect the global economy, particularly Western growth, says the BBC's Duncan Weldon. Read more: How worried should we be about Chinese share price falls? The BBC's Robert Peston says that in the short term, the world will have an increase in spending power, but over the long term, this would make some countries like the UK considerably poorer. Read more: Will China's slowdown make us poorer? But many anticipate that the Chinese government will continue to prop up the economy one way or the other, and even more so in light of this recent financial volatility. "It's difficult to see officials allowing the economy to slide further without some countervailing action," says Frederic Neumann, who co-heads Asian economics research at HSBC.

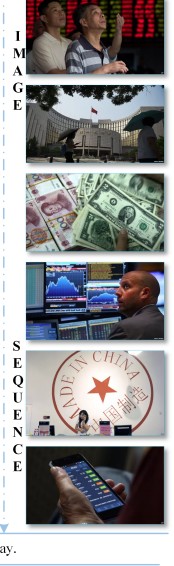

IMAGE SEQUENCE

| | |
|---|---|
| **English Summary** | **Ground Truth:** The repercussions from "Black Monday" - the global markets turmoil caused by a plunge in Chinese stocks - continue to be felt on Tuesday. |
| | **MMS:** Влияние « черного понедельника » - глобальных рыночных потрясений, вызванных падением китайского фондового рынка - продолжало проявляться во вторник. |
| | **MXLS:** چین کے ذریعے ایک دھوپ کی وجہ سے ہوتی ہے، سورج کے دن بھی احساس کیا جاتا ہے. |
| | **MMS+MXLS:** Konsekusi dari "Senin Hitam" - gangguan pasar global disebabkan oleh penurunan saham Cina - terus terasa pada Selasa. |
| | **Vanilla-KD:** The repercussions from "Black Monday" - the global markets turmoil caused by a plunge in Chinese stocks - continue to be felt on Tuesday. |
| | **D³TV (Ours):** Влияние « черного понедельника » - глобальных рыночных потрясений, вызванных падением китайского фондового рынка - продолжало проявляться во вторник. |
| **Russian Summary** | **Ground Truth:** Влияние « черного понедельника » - глобальных рыночных потрясений, вызванных падением китайского фондового рынка - продолжало проявляться во вторник. |
| | **MMS:** Konsekusi dari "Senin Hitam" - gangguan pasar global disebabkan oleh penurunan saham Cina - terus terasa pada Selasa. |
| | **MXLS:** The repercussions from "Black Monday" - the global markets turmoil caused by a plunge in Chinese stocks - continue to be felt on Tuesday. |
| | **MMS+MXLS:** Влияние « черного понедельника » - глобальных рыночных потрясений, вызванных падением китайского фондового рынка - продолжало проявляться во вторник. |
| | **Vanilla-KD:** چین کے ذریعے ایک دھوپ کی وجہ سے ہوتی ہے، سورج کے دن بھی احساس کیا جاتا ہے. |
| | **D³TV (Ours):** Konsekusi dari "Senin Hitam" - gangguan pasar global disebabkan oleh penurunan saham Cina - terus terasa pada Selasa. |
| **Urdu Summary** | **Ground Truth:** چین کے ذریعے ایک دھوپ کی وجہ سے ہوتی ہے، سورج کے دن بھی احساس کیا جاتا ہے. |
| | **MMS:** Влияние « черного понедельника » - глобальных рыночных потрясений, вызванных падением китайского фондового рынка - продолжало проявляться во вторник. |
| | **MXLS:** چین کے ذریعے ایک دھوپ کی وجہ سے ہوتی ہے، سورج کے دن بھی احساس کیا جاتا ہے. |
| | **MMS+MXLS:** Konsekusi dari "Senin Hitam" - gangguan pasar global disebabkan oleh penurunan saham Cina - terus terasa pada Selasa. |
| | **Vanilla-KD:** The repercussions from "Black Monday" - the global markets turmoil caused by a plunge in Chinese stocks - continue to be felt on Tuesday. |
| | **D³TV (Ours):** Влияние « черного понедельника » - глобальных рыночных потрясений, вызванных падением китайского фондового рынка - продолжало проявляться во вторник. |
| **Indonesian Summary** | **Ground Truth:** Konsekusi dari "Senin Hitam" - gangguan pasar global disebabkan oleh penurunan saham Cina - terus terasa pada Selasa. |
| | **MMS:** |
| | **MXLS:** The repercussions from "Black Monday" - the global markets turmoil caused by a plunge in Chinese stocks - continue to be felt on Tuesday. |
| | **MMS+MXLS:** Влияние « черного понедельника » - глобальных рыночных потрясений, вызванных падением китайского фондового рынка - продолжало проявляться во вторник. |
| | **Vanilla-KD:** چین کے ذریعے ایک دھوپ کی وجہ سے ہوتی ہے، سورج کے دن بھی احساس کیا جاتا ہے. |
| | **D³TV (Ours):** Konsekusi dari "Senin Hitam" - gangguan pasar global disebabkan oleh penurunan saham Cina - terus terasa pada Selasa. |

Figure 3: An example of many-to-many summarization in different language directions. ((updating later.))