# OpenReview forum: "D$^2$TV: Dual Knowledge Distillation and Target-oriented Vision Modeling for Many-to-Many Multimodal Summarization"
_EMNLP/2023/Conference — EMNLP 2023 Findings_

### Official Review · Reviewer_KUUr · 2023-08-03

**Soundness:** 3

**Excitement:**

3: Ambivalent: It has merits (e.g., it reports state-of-the-art results, the idea is nice), but there are key weaknesses (e.g., it describes incremental work), and it can significantly benefit from another round of revision. However, I won't object to accepting it if my co-reviewers champion it.

**Paper Topic And Main Contributions:**

This work is the first to introduce the comprehensive many-to-many multimodal summarization (M$^3$S) task and contribute a corresponding benchmark dataset. To achieve this, a dual knowledge distillation and target-oriented vision modeling framework (D2TV) is proposed.
Experiments on the M3Sum benchmark with four languages and 16 language directions show that this work builds new state-of-the-art performance, demonstrating the effectiveness of the D2TV approach.

**Reasons To Accept:**

1. This work introduces a new task called M$^3$S and presents a corresponding dataset.
2. The paper is well-written and easy to understand, making it accessible to a broad audience.

**Reasons To Reject:**

1. The motivation behind the Dual Knowledge Distillation module is unclear, and it would be preferable to provide vivid explanations with relevant examples in the Introduction section.
2. Figure 2 is not sufficiently clear, and the DKD and TCO modules are not presented in the MMS model. It is recommended to revise and modify them accordingly.
3. Recent relevant work is not be mentioned, such as "CFSum: A Coarse-to-Fine Contribution Network for Multimodal Summarization".

**Reproducibility:**

3: Could reproduce the results with some difficulty. The settings of parameters are underspecified or subjectively determined; the training/evaluation data are not widely available.

**Reviewer Confidence:**

2: Willing to defend my evaluation, but it is fairly likely that I missed some details, didn't understand some central points, or can't be sure about the novelty of the work.

---

> ### Author Rebuttal · Authors · 2023-08-29
>
> Dear Reviewer,
>
> We sincerely thank you for the careful reviews. These comments are all valuable and very helpful for revising and improving our work, as well as significant to guide our research. The comments are laid out below and specific concerns have been numbered. Our responses are given point by point. We have studied the comments carefully and made corresponding corrections. We hope the newly provided content could address your concerns or clarify some misunderstandings.
>
> With the strong support of our proposed M3Sum task and dataset by **Reviewer #2&#3&#4** and the strong backing of our novel approach by **Reviewer  #1&#2&#3**, we believe that this work has the potential to lay the foundation and facilitate future research in multimodal many-to-many abstractive summarization fields. We will release our dataset and codes with step-by-step instructions for reproducing our experiments.
>
> **Comment 1**: The motivation behind the Dual Knowledge Distillation module is unclear, and it would be preferable to provide vivid explanations with relevant examples in the Introduction section.
>
> **Response to Comment 1**: Essentially, the MXLS model needs to simultaneously conduct machine translation and summarization [1] while the MMS model only conducts summarization. Obviously, it is harder to train an MXLS model than to learn an MMS model and that is why researchers [2,3] take the MMS model as the teacher to help the MXLS student model. However, when the MXLS model achieves a level of multilingual and cross-lingual ability, the MXLS model can better transfer and share task knowledge among different languages. Therefore, the MXLS model, in turn, can guide the MMS model to conduct summarization in diverse languages (English->English, Indonesian->Indonesian, and Urdu->Urdu), especially for low-resource ones. That is why we propose DKD to mutually enhance their performance as we explained in Lines 239-259. We will follow your suggestions and add more detailed explanations in the Introduction part in the next version.
>
> [1] A survey on cross-lingual summarization. TACL 2022.
>
> [2] Improving neural cross-lingual abstractive summarization via employing optimal transport distance for knowledge distillation. AAAI 2021
>
> [3] Assist non-native viewers: Multimodal cross-lingual summarization for how2 videos. EMNLP 2022.
>
> **Comment 2**: Figure 2 is not sufficiently clear, and the DKD and TCO modules are not presented in the MMS model. It is recommended to revise and modify them accordingly.
>
> **Response to Comment 2**: Thanks for your insightful suggestions. We will revise our model architecture to clearly present the relationship between different modules.
>
> **Comment 3**: Recent relevant work is not be mentioned, such as "CFSum: A Coarse-to-Fine Contribution Network for Multimodal Summarization".
>
> **Response to Comment 3**: Because the submission deadline of EMNLP2023 is before the appearance of this work. Therefore, I cannot mention this work. I will discuss such close work in the new version.
> The work "CFSum: A Coarse-to-Fine Contribution Network for Multimodal Summarization" first appeared to Arxiv on July 06 2023 and the date of ACL is also July. However, the submission deadline for EMNLP is June 24, 2023.

---

### Official Review · Reviewer_CEcV · 2023-08-04

**Soundness:** 2

**Excitement:**

3: Ambivalent: It has merits (e.g., it reports state-of-the-art results, the idea is nice), but there are key weaknesses (e.g., it describes incremental work), and it can significantly benefit from another round of revision. However, I won't object to accepting it if my co-reviewers champion it.

**Paper Topic And Main Contributions:**

The article's main theme is "Many-to-Many Multimodal Summarization (M^{3}S)" task, which aims to generate summaries in any language using document inputs in any language and the corresponding image sequence, combining both multimodal monolingual summarization (MMS) and multimodal cross-lingual summarization (MXLS) tasks. The contribution of the paper can be summarized as follows:
Introduction of M^{3}S Task: The paper introduces the M^{3}S task as a general and practical problem, highlighting the lack of attention given to it in existing research.

Dual Knowledge Distillation and Target-Oriented Vision Modeling Framework: The framework utilizes a dual knowledge distillation method, leading to mutual improvement of both tasks. Additionally, a target-oriented contrastive objective is designed to provide target-oriented visual features, discarding irrelevant visual information.

Contribution of M^{3}Sum Dataset: The paper contributes a new dataset called "M^{3}Sum" for the M^{3}S task, which consists of 44 languages. This dataset is expected to facilitate future research in the field of many-to-many multimodal summarization.


**Questions For The Authors:**

A. What is the motivation for the proposed task and modules in your paper? Could you explain one of your important claims: “implicit learning via the MMS objective may limit the potential of visual features” in detail?

B. How does your model perform compared with prior baselines?

C. How does your model perform on settings containing more language, and is there any evidence to prove that your target-oriented contrastive objective can discard irrelevant visual information?


**Reasons To Accept:**

1. Significance: Given the increasing availability of multimodal data on the media. The M^{3}S task, which combines both multimodal monolingual summarization and multimodal cross-lingual summarization, offers a comprehensive and practical solution for generating summaries across multiple languages and modalities.

2. Novelty: The paper addresses a relatively under-explored problem in the field of natural language processing, namely the Many-to-Many Multimodal Summarization (M^{3}S) task. By introducing this task and proposing a new framework to tackle it, the paper brings fresh insights and contributions to the research community.

3. Experimental Results: The paper presents extensive experiments in a many-to-many setting, showcasing the certain effectiveness of the proposed approach. The inclusion of the newly contributed M^{3}Sum dataset with 44 languages further provides a resource for the community.


**Reasons To Reject:**

1. **Insufficiently Explained Motivation**：(a). The motivation to combine both MMS and MXLS is not well explained in the paper. While the paper mentions that there is little attention given to the M^{3}S task, it is not strong, it is better to provide a more convincing motivation. (b). The paper's rationale for using the explicit target-oriented contrastive objective instead of implicit learning via the MMS objective lacks sufficient explanation and depth. While the paper mentions that implicit learning may limit the potential of visual features, there is no comprehensive analysis or concrete evidence to support this claim. It is better to give a clear and detailed explanation of the limitations of implicit learning and how the target-oriented contrastive objective addresses those limitations.

2. **Insufficient Comparison with Existing Work**: The abstract mentions that little research has been done on the M^{3}S task, but it's essential for the paper to provide a more comprehensive comparison with existing related works in the field such as “Liu et al. (2022)” mentioned in line 054 in the paper. It is better to adequately situate itself in the existing literature and have a thorough comparison with state-of-the-art methods.

3. **Inadequate Evaluation on the M^{3}Sum Dataset**: While the paper contributes a new dataset called M^{3}Sum, however the evaluation on this dataset is only taken on 4 randomly selected languages, thus is insufficient and lacks diversity, it might raise doubts about the reliability of the results and the generalizability of the proposed approach. Moreover, the intuitional analysis of whether their Target-oriented Contrastive Objective discarding the irrelevant visual information is missing.


**Reproducibility:**

4: Could mostly reproduce the results, but there may be some variation because of sample variance or minor variations in their interpretation of the protocol or method.

**Reviewer Confidence:**

3: Pretty sure, but there's a chance I missed something. Although I have a good feel for this area in general, I did not carefully check the paper's details, e.g., the math, experimental design, or novelty.

---

> ### Author Rebuttal · Authors · 2023-08-29
>
> Dear Reviewer,
>
> We sincerely thank you for the careful reviews. These comments are all valuable and very helpful for revising and improving our work, as well as significant to guide our research. The comments are laid out below and specific concerns have been numbered. Our responses are given point by point. We have followed closely the suggestions and made clarifications and revisions accordingly. We hope the newly provided content could address your concerns or clarify some misunderstandings.
>
> With the strong support of our proposed M3Sum task and dataset by **Reviewer #2&#3&#4** and the strong backing of our novel approach by **Reviewer #1&#2&#3**, we believe that this work has the potential to lay the foundation and facilitate future research in multimodal many-to-many abstractive summarization fields. We will release our dataset and codes with step-by-step instructions for reproducing our experiments.
>
> **Comment 1**: Insufficiently Explained Motivation：(a). The motivation to combine both MMS and MXLS is not well explained in the paper. While the paper mentions that there is little attention given to the M^{3}S task, it is not strong, it is better to provide a more convincing motivation. (b). The paper's rationale for using the explicit target-oriented contrastive objective instead of implicit learning via the MMS objective lacks sufficient explanation and depth. While the paper mentions that implicit learning may limit the potential of visual features, there is no comprehensive analysis or concrete evidence to support this claim. It is better to give a clear and detailed explanation of the limitations of implicit learning and how the target-oriented contrastive objective addresses those limitations.
>
> **Response to Comment 1**: **(a)** Essentially, the MXLS model needs to simultaneously conduct machine translation and summarization [1] while the MMS model only conducts summarization. Obviously, it is harder to train an MXLS model than to learn an MMS model and that is why researchers [2,3] take the MMS model as the teacher to help the MXLS student model. However, when the MXLS model achieves a level of multilingual and cross-lingual ability, the MXLS model can better transfer and share task knowledge among different languages. Therefore, the MXLS model, in turn, can guide the MMS model to conduct summarization in diverse languages (English->English, Indonesian->Indonesian, and Urdu->Urdu), especially for low-resource ones. That is why we propose DKD to mutually enhance their performance as we explained in Lines 239-259. We will follow your constructive suggestions and add more detailed explanations in the Introduction part in the next version.
>
> **(b)** Actually, we follow this claim "implicit learning via the MMS objective may limit the potential of visual features" from [4]. For example, though the object “中国制造” in the fifth image of Fig. 1 is associated with the article content ("made in China"), the object contributes little to the summary. That is, the MAS model should focus on summary-oriented visual features. However, the visual features are generally implicitly learned via the MAS objective, which cannot help the model learn to explicitly discard such needless visual information.
> Therefore, we propose a target-oriented vision modeling objective to explicitly filter summary-unrelated visual features. The results show that the model with enhanced visual features achieves better results than vanilla visual features (Row 5 vs. Row 0 in Table 2), demonstrating the effectiveness of the proposed target-oriented vision modeling with faster training speed.
>
> [1]A survey on cross-lingual summarization. TACL 2022.
>
> [2] Improving neural cross-lingual abstractive summarization via employing optimal transport distance for knowledge distillation. AAAI 2021
>
> [3] Assist non-native viewers: Multimodal cross-lingual summarization for how2 videos. EMNLP 2022.
>
> [4] Summary-oriented vision modeling for multimodal abstractive summarization. ACL 2023. (appeared to Arxiv on Dec 15, 2022)
>
> **Comment 2**: Insufficient Comparison with Existing Work: The abstract mentions that little research has been done on the M^{3}S task, but it's essential for the paper to provide a more comprehensive comparison with existing related works in the field such as “Liu et al. (2022)” mentioned in line 054 in the paper. It is better to adequately situate itself in the existing literature and have a thorough comparison with state-of-the-art methods.
>
> **Response to Comment 2**: Actually, we have compared with the previously state-of-the-art model in [4]. The results are shown in Row 4 of Table 2, which shows that our model can significantly surpass it with faster training speed. Besides, we followed this constructive suggestion and have re-implemented the triple-stage training method of [5] as a stronger baseline. For example, the averaged results in terms of ROUGE-1 / ROUGE-2 / ROUGE-L / BERTSCORE scores are listed as follows:
>
>  Models | English->*|Indonesia->*|Russian->*|and Urdu->*
> -|-|-|-|-
> Liang et al. (2022)[4] | 30.19 / 10.57 / 24.03 / 68.45 |31.42 / 11.49 / 25.33 / 68.88 |30.75 / 10.59 / 24.41 / 68.64 |29.96 / 10.61 / 24.32 / 68.31
> Liu et al. (2022) [5] |29.88 / 10.36 / 23.78 / 68.33 | 31.09 / 11.60 / 24.83 / 68.83 |29.58 / 10.87 / 23.99 / 68.45 |29.63 / 10.82 / 23.53 / 68.12
> Ours |30.77 / 10.72 / 24.53 / 68.83 |32.08 / 11.71 / 25.63 / 69.09 |31.27 / 11.24 / 25.13 / 68.96 |30.70 / 10.91 / 24.71 / 68.75
>
> These results further show the effectiveness of our proposed approach. We will include these results in the revised version.
>
> [4]: Summary-oriented vision modeling for multimodal abstractive summarization. ACL 2023.
>
> [5]: Assist non-native viewers: Multimodal cross-lingual summarization for how2 videos. EMNLP 2022.
>
> **Comment 3**: Inadequate Evaluation on the M^{3}Sum Dataset: While the paper contributes a new dataset called M^{3}Sum, however the evaluation on this dataset is only taken on 4 randomly selected languages, thus is insufficient and lacks diversity, it might raise doubts about the reliability of the results and the generalizability of the proposed approach. Moreover, the intuitional analysis of whether their Target-oriented Contrastive Objective discarding the irrelevant visual information is missing.
>
> **Response to Comment 3**: (1) In existing work [1], it selects 4 languages to conduct experiments. Inspired by this work, we randomly chose 4 languages covering 16 language directions. (2) Another realistic reason is our limited GPU resources and space, we have to utilize 4 languages. We think the results in these directions have already verified our M3Sum dataset and the effectiveness of our model on the benchmark. Despite all this, we will follow your insightful suggestions and would like to include six languages in the appendix.
>
> (2) Row 5 vs. Row 4 in Table 2 shows that our target-oriented contrastive objective (TCO) can achieve a competitive performance to CAT[2] with faster training speed. The CAT [2] proposes two explicit auxiliary tasks including the vision to summary task and the masked image modeling task to discard irrelevant visual information, showing that they can filter target-unrelated visual features and obtain SOTA performance. Therefore, we believe that our method also can discard such irrelevant visual information and thus achieve a competitive performance to CAT.
>
> (3) Besides, we use the extracted visual features to reconstruct the original images (we aim to present which part of the image our model focuses on.). Taking Figure 1 as an example, the results show that the people, stock in the first image, and outline of money in the third image can be fuzzily obtained while the fifth image is nearly blurred. This shows that the model indeed can focus on the target-related part and ignore the irrelevant information to some extent. We will add more analysis about this in the revised version.
>
> [1] Models and Datasets for Cross-Lingual Summarisation. EMNLP2021.
>
> [2] Summary-oriented vision modeling for multimodal abstractive summarization. ACL 2023.
>
> **Comment 4**: What is the motivation for the proposed task and modules in your paper? Could you explain one of your important claims: “implicit learning via the MMS objective may limit the potential of visual features” in detail?
>
> **Response to Comment 4**: (a) Under the background of globalization and the rapid increase of multimedia data, there is still a lack of many-to-many multimodal summarization research, which may prevent diverse conversations and thus hinder the globalization process. This is why we propose the many-to-many multimodal summarization task. As for the modules of our architecture, we designed them from the perspective of the new task: (1) the dual knowledge distillation for many-to-many summarization (MMS and MXLS consist of many-to-many setting); (2) the target-oriented vision modeling to enhance the fusion of multimodality.
>
> (b) Actually, we follow this claim "implicit learning via the MMS objective may limit the potential of visual features" from [4]. For example, though the object “中国制造” in the fifth image of Fig. 1 is associated with the article content ("made in China"), the object contributes little to the summary. Thus, the MAS model should focus on summary-oriented visual features. However, the visual features are generally implicitly learned via the MAS objective, which cannot help the model learn to explicitly discard such needless visual information. Therefore, we propose an explicit summary-oriented training objective to filter target-unrelated visual features.
>
> [4]: Summary-oriented vision modeling for multimodal abstractive summarization. ACL 2023.
>
> **Comment 5**: B. How does your model perform compared with prior baselines?
>
> **Response to Comment 5**: Actually, besides the general baselines, e.g., MMS, MXLS, MMS+MXLS (multi-task learning), and vanilla-KD under the mT5 and mBART50 backbones, we have compared with the previously state-of-the-art model in [4]. The results are shown in Row 4 of Table 2, which shows that our model can significantly surpass it with faster training speed. We also have re-implemented the triple-stage training method of [5] as a stronger baseline. The averaged results in English->*, Indonesia->*, Russian->*, and Urdu->*  are 29.88 / 10.36 / 23.78 / 68.33, 31.09 / 11.60 / 24.83 / 68.83, 29.58 / 10.87 / 23.99 / 68.45, and 29.63 / 10.82 / 23.53 / 68.12, respectively. All these results of prior baselines further show the effectiveness of our proposed approach.
>
> [4]: Summary-oriented vision modeling for multimodal abstractive summarization. ACL 2023.
>
> [5]: Assist non-native viewers: Multimodal cross-lingual summarization for how2 videos. EMNLP 2022.
>
> **Comment 6**: How does your model perform on settings containing more language, and is there any evidence to prove that your target-oriented contrastive objective can discard irrelevant visual information?
>
> **Response to Comment 6**: (1) Row 5 vs. Row 4 in Table 2 shows that our target-oriented contrastive objective (TCO) can achieve a competitive performance to CAT [4] with faster training speed. The CAT [4] proposes two explicit auxiliary tasks including the vision to summary task and the masked image modeling task to discard irrelevant visual information, showing that they can filter target-unrelated visual features and obtain SOTA performance. Therefore, we believe that our method also can discard such irrelevant visual information and thus achieve a competitive performance to CAT.
>
> (2) Besides, we use the extracted visual features to reconstruct the original images (we aim to present which part of the image our model focuses on.). Taking Figure 1 as an example, the results show that the people, stock in the first image, and outline of money in the third image can be fuzzily obtained while the fifth image is nearly blurred. This shows that the model indeed can focus on the target-related part and ignore the irrelevant information to some extent.
>
> [4] Summary-oriented vision modeling for multimodal abstractive summarization. ACL 2023.

---

### Official Review · Reviewer_UzrF · 2023-08-10

**Soundness:** 4

**Excitement:**

4: Strong: This paper deepens the understanding of some phenomenon or lowers the barriers to an existing research direction.

**Paper Topic And Main Contributions:**

The authors introduce the Many-to-Many Multimodal Summarization (M3S) task, involving generating summaries across languages using document inputs and corresponding image sequences. Existing research has mostly focused on either monolingual summarization (MMS) or cross-lingual summarization (MXLS), neglecting M3S. This paper proposes a framework for M3S, combining knowledge distillation between MMS and MXLS and a target-oriented vision model to enhance both tasks. The dual knowledge distillation ensures knowledge transfer, while a target-oriented contrastive objective filters irrelevant visual features. Experimental results demonstrate the effectiveness of this approach, and a new dataset (M3Sum) with 44 languages is introduced for future research.

**Questions For The Authors:**

- The authors mention the blue results are MXLS results and the yellow is MMS results for Table 1. It was quite interesting to me that the MMS model did decently in the MXLS task while the MXLS model did much poorer for the MMS task. Why do the authors think this phenomenon occurs?
- It was also interesting to me that most models for the MXLS tasks outperformed MMS tasks. It seems to me that cross-lingual summarization may be more difficult than monolingual summarization. Do the authors have any incites to this?

**Reasons To Accept:**

- The paper is the first to introduce the Many-to-Many Multimodal Summarization (M3S) task and provides a novel benchmark dataset, M3Sum, to facilitate research in this area.
- The proposed Dual Knowledge Distillation and Target-Oriented Vision Modeling (D2TV) framework addresses key challenges in M3S. By enabling knowledge transfer between Monolingual Summarization (MMS) and Cross-Lingual Summarization (MXLS), and effectively filtering irrelevant visual information, the framework enhances the performance of multimodal summarization models.
- Empirical experiments conducted on the M3Sum benchmark demonstrate that the proposed D2TV framework achieves state-of-the-art performance. The model's ability to outperform existing methods across four languages and 16 language directions underscores the efficacy of the approach, reinforcing its potential contribution to advancing the field of multimodal summarization.

**Reasons To Reject:**

- Lack of results on languages. If there were some more results on a more variety of languages, it would be more enlightening to the performance of the framework.
- Although their framework is impressive, the improvement in performance compared to the other models (especially Vanilla-KD) seem to be minimal.

**Reproducibility:**

4: Could mostly reproduce the results, but there may be some variation because of sample variance or minor variations in their interpretation of the protocol or method.

**Reviewer Confidence:**

4: Quite sure. I tried to check the important points carefully. It's unlikely, though conceivable, that I missed something that should affect my ratings.

---

> ### Author Rebuttal · Authors · 2023-08-29
>
> Dear Reviewer,
>
> We sincerely thank you for the careful reviews. These comments are all valuable and very helpful for revising and improving our work, as well as significant to guide our research. The comments are laid out below and specific concerns have been numbered. Our responses are given point by point. We have followed closely the suggestions and made clarifications and revisions accordingly. We hope the newly provided content could help to further strengthen our work.
>
> With the strong support of our proposed M3Sum task and dataset by **Reviewer #2&#3&#4** and the strong backing of our novel approach by **Reviewer  #1&#2&#3**, we believe that this work has the potential to lay the foundation and facilitate future research in multimodal many-to-many abstractive summarization fields. We will release our dataset and codes with step-by-step instructions for reproducing our experiments.
>
> **Comment 1**: Lack of results on languages. If there were some more results on a more variety of languages, it would be more enlightening to the performance of the framework.
>
> **Response to Comment 1**: (1) In existing work [1], it selects 4 languages to conduct experiments. Inspired by this work, we randomly chose 4 languages covering 16 language directions.
> (2) Another realistic reason is our limited GPU resources and space, we have to utilize 4 languages. We think the results in these directions have already verified our M3Sum dataset and the effectiveness of our model on the benchmark. Despite all this, we will follow your insightful suggestions and would like to include six languages in the appendix.
>
> [1] Models and Datasets for Cross-Lingual Summarisation. EMNLP2021.
>
> **Comment 2**: Although their framework is impressive, the improvement in performance compared to the other models (especially Vanilla-KD) seem to be minimal.
>
> **Response to Comment 2**: In Table 1, under the mT5 backbone, our model significantly surpasses the Vanilla-KD method by about 1.0-1.5 ROUGE-L scores on average and 0.2-0.8 BERTScore on average. Under the mBART50 backbone, we also observe about 0.5-0.7 ROUGE-L gains and 0.3-0.5 BERTScore gains, which both are more significant than Vanilla-KD with t-test $p$ < 0.5.
>
> **Comment 3**: The authors mention the blue results are MXLS results and the yellow is MMS results for Table 1. It was quite interesting to me that the MMS model did decently in the MXLS task while the MXLS model did much poorer for the MMS task. Why do the authors think this phenomenon occurs?
>
> **Response to Comment 3**: We are sorry for this misunderstanding. For example, in the "English->" group under mT5 backbone, the MMS baseline works well (36.16 / 13.08 / 27.67 / 70.57) in English->English summary while performing worse in English-Indonesia (6.87 / 1.94 / 6.34 / 63.39), English-Russian (1.23 / 0.20 / 1.21 / 59.60), English ->Urdu (0.14 / 0.00 / 0.14 / 55.44) directions. For the MXLS baseline, it shows poor performance (6.94 / 2.35 / 6.03 / 61.8) in English-English direction while working better in cross-lingual ones (27.23 / 9.32 / 22.13 / 68.40 in English-Indonesia; 22.52 / 7.88 / 18.07 / 64.84 in English-Russian; 32.27 / 11.17 / 25.15 / 68.29 in English-Urdu). In all cases, we can find a similar and formal pattern.
>
> Except for the Russian-> Indonesia direction under the mBART50 backbone, the MXLS achieves slightly worse performance than MMS. The reason may be that (1) the original mBART50 has been trained on multilingual machine translation parallel data and owns a better cross-lingual ability that only learns the summarization ability in MMS model; (2) At MMS training, the monolingual summarization in four languages (English->English, Russian->Russian, Indonesia->Indonesia, Urdu->Urdu) is jointly trained which may enhance the summarization for different languages.
> We will add more clear analysis to clarify this in the next version.
>
> **Comment 4**: It was also interesting to me that most models for the MXLS tasks outperformed MMS tasks. It seems to me that cross-lingual summarization may be more difficult than monolingual summarization. Do the authors have any incites to this?
>
> **Response to Comment 4**: We are sorry for this misunderstanding. The results out of each block (e.g., English->English block) cannot be compared to others (e.g., Indonesia->English block) because they belong to different language directions. Therefore, in each block of MMS, the MMS always surpasses MXLS without any exception. In each block of MXLS, the MXLS always surpasses MMS without any exception. We will make these clearer in the next version.

---

### Official Review · Reviewer_16rq · 2023-08-11

**Soundness:** 3

**Excitement:**

3: Ambivalent: It has merits (e.g., it reports state-of-the-art results, the idea is nice), but there are key weaknesses (e.g., it describes incremental work), and it can significantly benefit from another round of revision. However, I won't object to accepting it if my co-reviewers champion it.

**Paper Topic And Main Contributions:**

This paper proposes a dual knowledge distillation and target-oriented vision modeling framework for many-to-many multimodal summarization (M3S). The authors demonstrated that this framework is better than multimodal monolingual summarization (MMS) and cross-lingual summarization (MXLS), MMS-MXLS, and vanilla knowledge distillation baselines. The experiments were conducted on the M3Sum dataset that consists of forty-four languages, in which the authors use four languages, namely English, Indonesian, Russian, and Urdu. Urdu is a low-resource language. The authors also conducted an ablation study based on mT5 and showed the usefulness of each feature in improving overall results for 4x4 languages. A small-scale human evaluation was also conducted for English --> English and Russian --> English test sets by three Chinese postgraduate students who majored in English. Details about the dataset, training computer, and one qualitative example are shown in the appendix.

**Questions For The Authors:**

Does the order of the image sequence matter?

**Reasons To Accept:**

The proposed framework (details in Fig. 2) demonstrates the capabilities to leverage the interconnection between MMS and MXLS through dual knowledge distillation to improve the overall performance. The paper is well-written and easy to follow, even with the detailed formulas. The experiments are thorough and well-presented, including both automated and human evaluations.

**Reasons To Reject:**

Table 2 explicitly shows that visual features are insignificant in the summarization task. Therefore, comparing the proposed method with text-only summarization baselines will be meaningful, especially with the recent strong Large Language Models, which multilingual solid supports.

**Reproducibility:**

4: Could mostly reproduce the results, but there may be some variation because of sample variance or minor variations in their interpretation of the protocol or method.

**Reviewer Confidence:**

4: Quite sure. I tried to check the important points carefully. It's unlikely, though conceivable, that I missed something that should affect my ratings.

---

> ### Author Rebuttal · Authors · 2023-08-29
>
> Dear Reviewer,
>
> We sincerely thank you for the careful reviews. These comments are all valuable and very helpful for revising and improving our work, as well as significant to guide our research. The comments are laid out below and specific concerns have been numbered. Our responses are given point by point. We have studied the comments carefully and made corresponding corrections. We hope the newly provided content could address your concerns or clarify some misunderstandings.
>
> With the strong support of the proposed M3Sum task and dataset by **Reviewer #2&#3&#4** and the strong backing of our novel approach by **Reviewer  #1&#2&#3**, we believe that this work has the potential to lay the foundation and facilitate future research in multimodal many-to-many abstractive summarization fields. We will release our dataset and codes with step-by-step instructions for reproducing our experiments.
>
> **Comment 1**: Table 2 explicitly shows that visual features are insignificant in the summarization task. Therefore, comparing the proposed method with text-only summarization baselines will be meaningful, especially with the recent strong Large Language Models, which multilingual solid supports.
>
> **Response to Comment 1**: Actually, the averaged results in four language directions of Row 1 are lower (0.3-0.5 ROUGE-L Scores on average) than Row 0 in Table 2, showing the importance of incorporating visual features for the summarization task. In our experiments (based on mT5 backbone), the average results of text-only summarization baseline are 28.48 / 9.44 / 22.73 / 67.71 in English->* directions, which is lower than text-vision-based ones (30.77 / 10.72 / 24.53 / 68.83), demonstrating the necessity of visual features. Therefore, we did not list detailed experiments of text-only baselines in Table 1 due to space limitations.  Following your insightful suggestions, we will add these experiments in Table 1 in the updated version.
>
> **Comment 2**: Does the order of the image sequence matter?
>
> **Responese to Comment 2**: Yes. In our early experiments, we found that the order has a positive impact on the model performance because it introduces additional temporal information. It is consistent with previous work [1,2].
>
> [1] Multi-modal summarization for asynchronous collection of text, image, audio and video. EMNLP 2017.
>
> [2] Summary-oriented vision modeling for multimodal abstractive summarization. ACL 2023.

---

### Meta-Review · Area_Chair_Arht · 2023-09-19

**Recommendation:** 4

**Metareview:**

The paper brings merits to Many-to-Many Multimodal Summarization, which is interesting. The model part is also reasonable. The major weakness is marginal results make it hard to justify how much vision can contribute to this task.

---

### Decision · Program_Chairs · 2023-10-07

**Decision:**

Accept-Findings

**Comment:**

The paper brings merits to Many-to-Many Multimodal Summarization, which is interesting. The model part is also reasonable. The major weakness is marginal results make it hard to justify how much vision can contribute to this task.